# Stigma towards Mental Disorders among Nursing Students and Professionals: A Bibliometric Analysis

**DOI:** 10.3390/ijerph19031839

**Published:** 2022-02-06

**Authors:** Concepción Martínez-Martínez, Francisca Esteve-Claramunt, Blanca Prieto-Callejero, Juan Diego Ramos-Pichardo

**Affiliations:** 1Department of Nursing, Faculty of Health Sciences, Universidad Europea de Valencia, 46010 Valencia, Spain; concepcion.martinez@universidadeuropea.es; 2Nursing Department, University of Huelva, 21004 Huelva, Spain; blanca.prieto@denf.uhu.es (B.P.-C.); juan.ramos@denf.uhu.es (J.D.R.-P.); 3Hospital Virgen de la Bella (Lepe), 21440 Huelva, Spain

**Keywords:** mental disorders, nursing, stigma, lived-experience, bibliometric analysis

## Abstract

Stigma is one of the main barriers to prevention, treatment and recovery from mental illness. However, bibliometric studies in this area are still scarce. Therefore, our aim was to quantify and analyze the scientific literature on the stigma of nursing students and professionals towards mental disorders. To this purpose, bibliometric indicators of scientific production, impact and collaboration were used. Among our results, it stands out that only 14.3% of the total number of studies analyzed measure the efficacy of the interventions carried out to reduce stigma. Furthermore, with exceptions such as Happell B and Byrne L, collaborations between authors and institutions are limited. “Service user involvement” appeared as a prominent keyword in 2018, coinciding with the increase in publications on the effectiveness of interventions. Interventions based on the involvement of people with psychiatric diagnoses in the design of nursing curricula seem to become a promising line of research. More studies measuring the efficacy of such interventions are needed. Knowledge of the lines of research that are being developed and of the researchers and institutions involved can contribute to creating synergy between the different researchers and to continue adding projects to the existing ones, thus contributing to the generation of more robust results that show the most indicated interventions to reduce the still present stigma and improve care for people with psychiatric diagnoses.

## 1. Introduction

Although mental health is a priority for the World Health Organization (WHO), as expressed by its statement that there can be “no health without mental health” [1], mental disorders remain one of the main causes of disability and dependence. In 2007, neuropsychiatric diseases accounted for 14% of the global burden of diseases [2]. Moreover, their incidence continues to increase annually, such that the WHO estimates that by 2030, depression will be the leading cause of disability worldwide. The economic burden attributed to mortality and morbidity associated with mental illness exceeds 4% of the gross domestic product (GDP), approximately EUR 600 billion, of the 28 countries that comprise the European Union [2]. Several national governments are implementing initiatives to reduce the impact of mental illness and its associated costs. However, the expected results have not been achieved. There is still a large difference between the need for treatment by users and its provision by healthcare systems. In low- and middle-income countries, between 76% and 85% of individuals with severe mental disorders do not receive treatment, and in high-income countries, the figure is between 35% and 50% [1].

Stigma is one of the main barriers to the prevention, treatment of and recovery from mental illness. Unsurprisingly, because of its negative consequences, stigma is considered by several researchers to be a “second disease” for individuals with a diagnosed mental disorder [3,4]. Stigma occurs when negative attributes, such as dangerousness and lack of responsibility and credibility, are associated with a person or group. Consequently, there is a desire for distancing and avoidance on the part of the majority social group, which results in the stigmatized group’s loss of status and discrimination at both the individual and structural levels [5]. In the field of health services, health professionals exhibit stigmatizing behaviors towards individuals with mental illness [6]. Health professionals’ perceptions of schizophrenia, depression and substance abuse do not differ from the views held by the general population [6,7,8]. Specifically, nursing professionals perceive individuals with mental illness as dangerous, unpredictable and emotionally unstable and experience fear, guilt and hostility towards patients with psychiatric illnesses [8]. Because of the stereotypical association with danger, health professionals are less willing to care for the patient or to administer such care alone [9,10]. In turn, the lack of credibility afforded to persons with mental disorders makes it possible for discomfort described by the patient to be attributed to side effects of psychotropic drugs or to the mental disorder itself rather than to the presence of a somatic pathology, resulting in fewer preventive interventions [11,12,13]. Consequently, individuals diagnosed with schizophrenia or bipolar disorder have a higher risk of high blood pressure, diabetes or cardiac or respiratory problems than the general population and have a shorter life expectancy due to the associated complications [14].

Scientific journals are the main means of validating new knowledge and disseminating it within the research community and to society and its agents [15]. Bibliometric analyses are increasingly used in the medical sciences because they facilitate the evaluation of published scientific research results and analysis of both scientific productivity on a given topic and the impact of research on a specific topic in terms of visibility, trends and future inquiry. However, few such bibliometric studies were conducted in the field of nursing and psychiatry [16,17], although nursing is one of the health professions that can make the greatest contribution to the treatment and recovery of individuals suffering from mental illness.

In this study, we used bibliometric indicators of scientific production, impact and collaboration with the aim of quantifying and analyzing the scientific literature on the stigma of nursing students and professionals towards mental disorders. Specifically to know the research activity and collaboration among authors and identify future trends in this line of research.

## 2. Materials and Methods

### 2.1. Studied Databases

In accordance with other similar studies [18,19], the Web of Science (WOS) platform was used to conduct this study. The WOS research engine is commonly used in studies that analyze scientific activity due to its multidisciplinary nature. It covers all scientific and technological fields, including more than 12,000 journals worldwide that correspond to more than 150 disciplines, and includes information on the citations generated and the addresses of all institutions [20].

Specifically, the articles for this study were extracted from five databases consulted through the Web of Science (WOS) platform: the Web of Science Core Collection, Current Contents Connect, Medline, SciELO Citation Index and the Korean Journal Database.

### 2.2. Search Profile and Information Downloading

To identify the maximum number of studies, all terms in the title or abstract were searched. The search equation was as follows: (Nurse [Title OR Abstract] OR Nursing [Title OR Abstract]) AND (Stigma [Title OR Abstract] OR Prejudice [Title OR Abstract] OR Attitude [Title OR Abstract]) AND (Mental Illness [Title OR Abstract] OR Mental Disorder [Title OR Abstract]).

Only the document type filter was applied, excluding letters, books, biographies, reference material, abstracts, meetings and news or corrections. No filter was applied in the field “years of publication”. Thus, all documents published up to December 2019 were retrieved. Figure 1 show the flow diagram based on PRISMA guidelines [21].

### 2.3. Inclusion and Exclusion Criteria

The inclusion criteria were that the studies be carried out with nursing professionals and/or students, that stigma towards the mental disorder be analyzed or that an intervention be carried out to reduce said stigma. All studies were included regardless of study type and language. Studies with mixed samples of nurses and/or nursing students with other health professionals, as well as studies that address stigma towards physical illnesses, were excluded.

### 2.4. Identification of Relevant Studies and Data Extraction

To increase the validity, reliability and rigor of the study, two researchers independently read the title and abstract of the articles retrieved through the search strategy and applied the inclusion and exclusion criteria. The obtained results were compared. Disagreements were resolved by the two researchers reading the full articles and reaching a consensus.

### 2.5. Bibliometric Indicators

To achieve the proposed objectives, the following three types of bibliometric indicators were used.

Scientific productivity indicators: The authors, institutions, journals, most productive countries, types of published documents and language of publication were analyzed.

Impact indicators: The indicators used to evaluate the scientific impact of the retrieved data were journal impact factor, citations received and journal rank in the thematic area in which the journal was included. These data were obtained from the Science Citation Index Expanded (SCIE) and Social Science Citation Index (SSCI) from the WOS.

Collaboration indicators: The co-authorship and co-citation networks were analyzed, and the co-occurrence of authors’ keywords was visualized.

### 2.6. Data Analysis

The data retrieved from the WOS were linked to Microsoft Excel 19.0 software to calculate percentages and frequencies and to create Figure 2 and all tables. The Excel spreadsheets, in turn, were linked to Acces version 16.0 software to create reports and queries to analyze the data and obtain the bibliometric indicators. Reports and queries were created in order to analyze bibliometric indicators and obtain global data. The data on authors, citations and keywords were analyzed with VOSviewer software version 1.6.14. to obtain the bibliometric networks and visualizations. VOSviewer applies the association strength normalization technique, the VOS (visualization of similarities) technique and the clustering technique. Institutions and countries were analyzed with ArcGIS 10.1 software to create the geographical distribution map according to their productivity.

### 2.7. Ethical Considerations

The study did not involve research on humans or animals, nor did it include human participants. Therefore, the study did not require the approval of an ethics committee or informed consent.

## 3. Results

### 3.1. Evolution of Publications by Year

Based on the search strategy, a total of 1619 documents were obtained. After inclusion and exclusion criteria were applied, 195 were selected for the final part of this study because they fit the study topic. The selected articles were published between 1966 and 2019. Figure 2 show the temporal evolution of scientific production. The number of studies published annually on this topic was low and remained practically constant until 2005, after which the number of publications began to increase, achieving a highly significant increase in the final five years.

### 3.2. Document Types and Languages

Regarding the typology of the retrieved studies, 153 (78.97%) were observational studies, 29 (14.35%) were studies measuring the effectiveness of interventions to reduce stigma held by nursing students and professionals towards persons with severe mental disorders and 11 (5.64%) were literature reviews (Figure 3).

The studies were written in six different languages, with a clear predominance of English (*n* = 180; 92.3%), followed by Korean (*n* = 10; 5.12%).

### 3.3. Top Authors

Of the 547 different authors who published a study in the subject area, 477 (87.2%) published a single study. In contrast to the group of transitory authors, another group consisting of stable, more productive authors emerged, of whom 69 (12.6%) published between 2 and 10 studies, with only 1 author (0.18%) having published more than 10 studies. The two authors with the most studies were Brenda Happell (*n* = 19) and Chris Platania-Phung (*n* = 9). Table 1 show the authors who published five or more studies on stigma toward mental illness among nursing students and professionals.

### 3.4. Institutions and Countries with the Most Publications

The selected studies were published by 248 institutions from 46 countries. The countries with the most studies published were the United States and Australia, with 31 studies, followed by England with 17 (Figure 4).

When the countries in which the 195 selected studies were conducted were classified according to the Socio-Demographic Index (SDI) values [22], most of them were found to have a High SDI (152 countries), corresponding to 77.94% of the total, followed by 23 (11.79%) with a Medium SDI, 12 (6.15%) with a Medium-Low SDI, 7 (3.85%) with a Medium-High SDI and only 1 country (0.51%) with a Low SDI (see Appendix A).

The institutions to which the authors who published the most belonged were Central Queensland University and the University of Canberra, both in Australia, and the University of Turku (Finland) (Table 2).

### 3.5. Top Journals

The reviewed studies on stigma towards mental disorders among nursing students and professionals were published in a total of 86 journals, although the majority of articles (*n* = 104) (73.80%) were published in 10 journals (Table 3). The top-ranking was occupied by two journals, both with 22 published studies: the Journal of Psychiatric and Mental Health Nursing (JPMHN), with 589 citations received and the International Journal of Mental Health Nursing (IJMHN), with 543 citations received. These two were followed by two journals with 12 published studies each: the Archives of Psychiatric Nursing, with 296 citations, and the Journal of Advanced Nursing, with 271 citations received.

### 3.6. Co-Authorship, Co-Citations and Keyword Co-Occurrence Analysis

#### 3.6.1. Co-Authorship

The collaboration network between authors was obtained through VOSviewer. The visualization scale adopted was developed according to the individual productivity of each researcher (Scale = “Documents”) and, as a normalization method, the method of strong links was used (Method = “Strength of association”). Each node represents an author.

A single cluster appears made up of researchers with five or more documents published per year. Of the total number of authors of the articles analyzed, these are 19 authors who present scientific collaborations in a relatively stable way. The total strength of the links of most of them is similar (108-88) (Table 1). Greater size and proximity between the nodes representing Berta Happell and Cris Platania Phung is observed (Figure 5).

#### 3.6.2. Co-Citations

When the filter showing authors with a minimum of 10 citations was applied, a set of 44 authors was obtained. The total strength of the co-citation links with the other authors was calculated. The total link strength among these authors was 4.417. The authors with the highest number of co-citations were Corrigan P.W. and Happell B. Figure 6 present the four obtained clusters. Cluster 1 (red) covers 21 items, including Corrigan P.W. with a total link strength of 879, Link B.G. with 423, Thornicroft G. with 395 and Angermeyer M.C. with 342. Cluster 2 (green) has 11 items, and its most-cited authors are Happell B. with 1682, Byrne I. with 399 and Henderson, with 199. Cluster 3 (blue) has three elements, with the most co-citated author being Kassam A. with 178. Cluster 4 (yellow) includes Ross C. as the most co-cited author, with a total link strength of 399.

#### 3.6.3. Keyword Co-Occurrence

The 485 keywords were grouped according to the strength of occurrences. The keywords with the most co-occurrences were “attitude”, with 74, “mental disorders”, with 69, “stigma”, with 50 and “nurse”, with 40. Observing the variation in keywords by year revealed that in the final 10 years, “stigma” and “nursing students” were the most commonly used keywords, and since 2017, the keywords “consumer participation” and “health-related stigma” have been predominantly used. “Service user involvement” appeared as a prominent keyword in 2018 (Figure 7).

## 4. Discussion

The aim of this study has been to synthesize information on the production, impact, areas of interest and general characteristics of studies conducted on the stigma of nursing students and professionals towards people with mental disorders

The earliest article related to nursing and attitudes towards mental illness that was identified was Change in nursing students’ expectations regarding mental patients [23]; this work was published in the *US Journal of Nursing Research*. After 1966, the production trend remained stable, with an average of one article published per year, until 1996, a year in which production began to increase gradually. Between 1999 and 2005, the European Commission published a series of resolutions and conclusions about the need to promote mental health and combat stigma and discrimination towards those suffering from mental illness [24]. In addition, the WHO published a document characterizing the magnitude of the problem in terms of the number of people affected by a mental disorder and its future repercussions [25]. We believe that the resolutions adopted by the European Commission, together with the WHO’s declaration, contributed decisively to an increase in the number of studies on this topic, with the most-cited article thus far having been published in 2009: Stigma, negative attitudes and discrimination towards mental illness within the nursing profession: A review of the literature [12]. This article was published in the *JPMHN*, and it has a total of 168 citations for an average of 14 citations per year. The year 2015 marked the acceleration of the number of publications, with 86 articles published in just 5 years, representing 44.10% of the total production.

Our results show that the countries that have published the most on this topic are the USA (*n* = 29), Australia (*n* = 27) and England (*n* = 17). These results are in line with those obtained by [26]. The USA traditionally leads lists of countries with the most publications. This result is partly expected due to the country’s large number of research institutions and universities; thus, in absolute numbers, its number of publications is very high. Australia has increased its production of nursing publications since 2000 due in part to two factors: First, since that date, nursing professionals are able to choose to earn a doctorate; and second, Australian institutions have shown an interest in making research results visible with the aim of improving clinical practice [27]. In fact, the two authors who have researched the study topic the most, Brenda Happell and Chris Platania-Phung, are affiliated with Australian institutions.

On the other hand, it should be noted that 77.9% of the studies were conducted in countries with high SDI. The SDI is an index that measures the socio-economic development of a region and is therefore closely related to health outcomes. It is in these countries that the issue of professional stigma is of interest and studied. It seems logical for several reasons. One of them is, as we have indicated above, developed countries conduct the most gross research (on any topic), and that is reflected in this topic as well. However, other reasons could be that in less developed countries, research efforts are focused on health problems that are more significant for them or where nurses are still minorly involved in health research, or it may even be related to cultural and/or organizational issues related to the view of mental illness or the scarcity of specialized services to care for people with mental illness.

Analysis of a co-citation map reveals the most influential authors on a specific topic and the closeness of the research topics [28]. We identified two main clusters. The largest cluster is formed mainly by authors affiliated with large research groups. In particular, Corrigan P.W. and Link B.G. are psychologists affiliated with research groups on stigma in the US. Thornicroft G. and Angermeyer M.C. are psychiatrists who belong to research groups on stigma in the UK and Germany, respectively. These authors have researched stigma since the 1990s and have investigated the stigma process and the consequences for persons suffering from stigma. In addition, they have developed a theoretical framework to generate an operational definition and allow for the quantification of this phenomenon [5,29]. The most co-cited author in this study, Brenda Happell, belongs to the next largest cluster. This author has collaborated in research groups from various Australian universities, with one of her main lines of research being lived-experience participation in nurse education [30].

At the author level, analysis of the co-authorship graph reveals that most studies were carried out in isolation, with little cooperation among authors from different entities and countries. For example, authors from Australia, China, Spain, the UK and the US published studies on the effectiveness of different interventions to reduce stigma in nursing students and professionals, but no collaboration was observed among them [31,32,33,34,35]. One of the exceptions to this phenomenon of localism in research was found in a group of authors who have collaborated frequently on the subject of interest. Happell B. and Byrne L. (two of the most co-cited authors according to our results), together with authors belonging to other institutions (Platania-Phung Ch., Schulz B., Bocking J. and Bradshaw J., among others), have collaboratively published various studies on lived-experience participation in nurse education [30,36,37,38,39].

Possibly due to the influence of these collaborations, we observed the evolution of the co-occurrence of keywords over the last 10 years. Synchrony was observed between the greater visibility of words related to nursing students and mental health service users and the increase in the publication of studies on lived-experience participation in nurse education. Moreover, in 2018, coinciding with greater use of the keywords “service user involvement” and “consumer participation”, the COMMUNE project (Co-production of Mental Health Nursing Education) was published [40]. We believe that a line of research with experienced stakeholders developed by the authors mentioned above has been well established via a collaborative project at the international level. This project is being developed in five countries: Australia, Finland, Iceland, Ireland and the Netherlands. Mental health nurses, together with users of mental health services, are collaborating closely in the development of each of the project phases with the objective of implementing a “mental health recovery” module within the nursing degree curriculum [41,42].

When reviewing the nature of the studies included in this work and noting that most are observational, one can speculate on a change in trends. Starting in 2010 and coinciding again with the emergence of new lines of research involving people with a psychiatric diagnosis, the number of experimental studies measuring the effectiveness of anti-stigma interventions increased [34,43,44,45]. These results are in line with those obtained by Heim et al. (2019) [46], who also observed an increase in intervention studies in recent years.

Stigma is known to be one of the main barriers to maintaining physical health and to recovery in people with mental disorders [47,48]. Although research provides evidence for this theory, the data from several reviews indicate that nursing students and professionals maintain stigmatizing attitudes towards people with mental disorders [8,12,49]. We believe that continuing the lines of research initiated in recent years, in which attempts are being made to implement systematized training interventions with the participation of experts by experience within the official curricula for earning a nursing degree, will contribute to reducing stigmatizing attitudes among future nurses in addition to improving care for people with a psychiatric diagnosis.

### Limitations

This study has several limitations that must be recognized. First, a single search engine, the WOS, was used, which could bias the results. Second, due to the overlap of publications, the creation of research networks, self-citations and the mobility of researchers from one institution to another, a careful interpretation of the list of active authors and institutions may be necessary.

On the other hand, in this study, the institutional attribution of documents and authors was based on the total count criterion. That is, each document was assigned simultaneously to each of the institutions affiliated with the authors. Although there were other options, this attribution was chosen because it allowed the participation of the different institutions to be quantified separately (although it had the disadvantage of increasing the number of documents in the counts).

## 5. Conclusions

To our knowledge, this is the first study to perform a bibliometric analysis of the publications available to date on the stigma of nursing students and professionals towards people with mental disorders. It was observed that in recent years there was an increase in research articles on this topic; however, only 15% of the texts identified were aimed at designing and measuring the effectiveness of interventions to reduce it. Currently, the most promising lines of research are those that design interventions involving people with lived experience. In this regard, in recent years, an international project has emerged in which experienced persons collaborate in the design and implementation of a recovery module that is included in nursing curricula [39]. Despite this, it appears that this international collaboration is the exception, as most research is conducted in very small contexts, with national collaborations at most.

This study has made it possible to identify the main authors and institutions that research and publish on the stigma of nursing professionals towards people with mental disorders, as well as to learn about new lines of research. We believe that making this information known through this article can help to create synergies and to develop new collaborative projects for the existing ones. Collaboration between different authors, institutions and countries, especially between researchers and institutions from “high” and “low” or “middle” SDI countries, is essential to generate more robust results that show which are the most appropriate interventions to reduce the stigma still present and improve care for people with psychiatric diagnoses.

## Figures and Tables

**Figure 1 ijerph-19-01839-f001:**
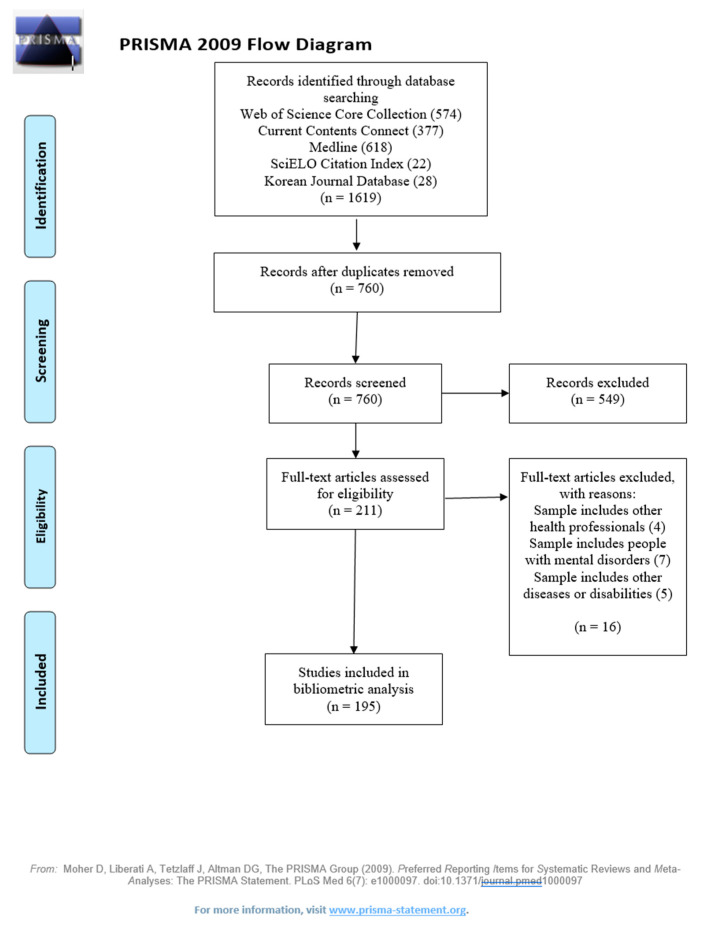
Information flow of the process followed for paper selection.

**Figure 2 ijerph-19-01839-f002:**
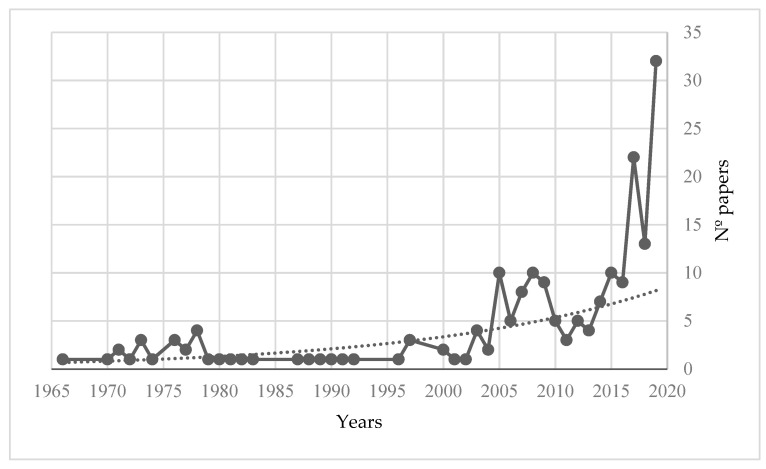
Number of articles published per year.

**Figure 3 ijerph-19-01839-f003:**
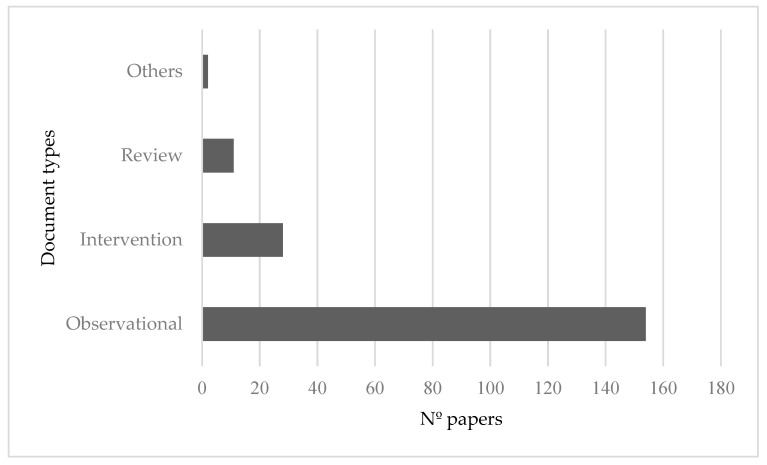
Number of articles according to the type of research design used.

**Figure 4 ijerph-19-01839-f004:**
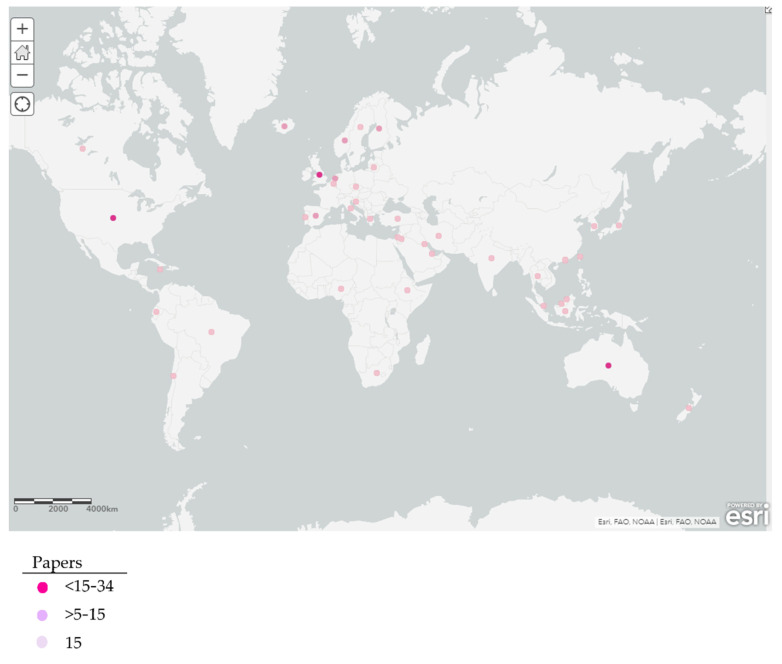
Map of the most productive countries obtained according to institutional affiliation.

**Figure 5 ijerph-19-01839-f005:**
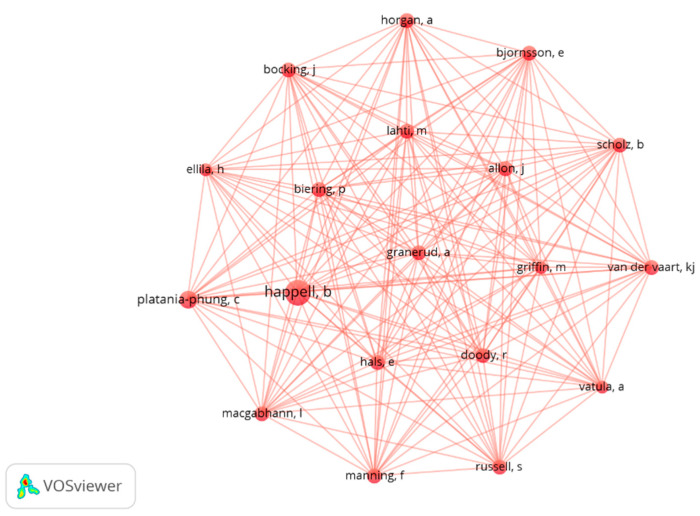
Co-authorship network.

**Figure 6 ijerph-19-01839-f006:**
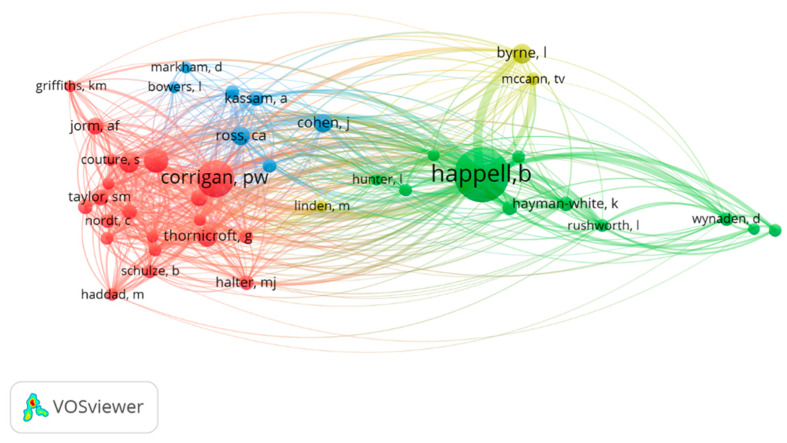
Author co-citation map.

**Figure 7 ijerph-19-01839-f007:**
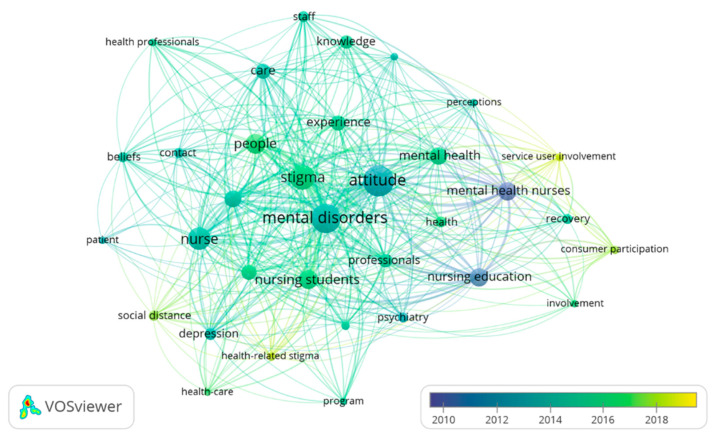
Author keywords co-occurrence network.

**Table 1 ijerph-19-01839-t001:** Most productive authors with five or more published works.

Author	Institutional Affiliation	No. Documents	Total Link Strenght
Happell, B	Univ Newcastle, Sch Nursing & Midwifery	19	108
Platania-Phung, Ch	Univ Newcastle, Sch Nursing & Midwifery	9	108
Allon, J	Univ Appl Sci Utrecht, Inst Nursing Studies	6	105
Biering, P	Iceland Univ, Dept Nursing	6	105
Bjornsson, E	Iceland Univ, Dept Nursing	6	105
Bocking, J	Canberra Univ, Fac Hlth, Sch Hlth Sci,	6	105
Doody, R	Univ Coll Cork, Sch Nursing & Midwifery	6	105
Granerud, A	Inland Norway Univ Appl Sci, Fac Hlth & Social Sci	6	105
Hals, E	Inland Norway Univ Appl Sci, Fac Hlth & Social Sci	6	105
Horgan, A	Univ Coll Cork, Sch Nursing & Midwifery	6	105
Lahti, M	Univ Turku Appl Sci, Health & Well Being	6	105
MacGabhann, L	Dublin City Univ, Sch Nursing & Human Sci	6	105
Manning, F	Univ Coll Cork, Sch Nursing & Midwifery	6	105
Russell, S	Dublin City Univ, Sch Nursing & Human Sci	6	105
Scholz, B	Australian Natl Univ, ANU Med Sch, Coll Hlth & Med	6	105
Van Der Vaart, KJ	Faculty of Science, Leiden University, Leiden	6	105
Badamath, S	National Institute of Mental Health & Neurosciences, India	5	105
Ellila, H	Univ Turku Appl Sci, Health & Well Being	5	88
Griffin, M	Dublin City Univ, Sch Nursing & Human Sci	5	88
Vatula, A	Univ Turku Appl Sci, Health & Well Being	5	88

**Table 2 ijerph-19-01839-t002:** Ranking of institutions with the most published authors.

Institutions	Documents	Country
Central Queensland University	8	Australia
University of Canberra	8	Australia
University of Turku	8	Finland
Dublin City University	7	Ireland
Australian National University	6	Australia
Inland Norway University of Applied Sciences	6	Norway
Univ Appl Sci Utrecht	6	Netherlands
University College Cork	6	Ireland
University of Iceland	6	Iceland
University of Newcastle	6	Australia
National Institute of Mental Health	5	USA
University of London	5	England
University of Melbourne	5	Australia

**Table 3 ijerph-19-01839-t003:** Top published journals on stigma towards mental disorder among nursing students and professionals.

Journals	No. of Papers	%	^†^ IF	Total Cites	Cites by Document
International Journal of Mental Health Nursing	22	11.30	2.433	543	24.68
Journal of Psychiatric and Mental Health Nursing	22	11.30	2.009	589	26.77
Archives of Psychiatric Nursing	12	6.15	1.299	296	24.66
Journal of Advanced Nursing	12	6.15	2.376	271	22.50
Issues In Mental Health Nursing	10	5.12	0.977	221	22.10
Journal of Psychosocial Nursing and Mental Health Services	7	3.59	0.710	93	9.80
Nursing Research	6	3.07	2.020	29	4.80
Journal of Clinical Nursing	5	2.56	1.757	22	4.40
International Journal of Nursing Studies	4	2.05	3.570	140	3.50
Investigación y Educación en Enfermería	4	2.05	0.250	8	2

^†^ IF = Impact Factor.

## Data Availability

The original contributions presented in the study are included in the article/Appendix A, further inquiries can be directed to the corresponding author.

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
