# Peer review of "Stigma towards Mental Disorders among Nursing Students and Professionals: A Bibliometric Analysis"

_ijerph, 2022, doi:10.3390/ijerph19031839_

Round 1
Reviewer 1 Report
Many thanks for giving me the chance to review this interesting and innovative article that aimed to evaluate the volume of the literature about stigma towards mental disorders held by students and nursing professionals.
Abstract:
- Please add more details about the methods because is not clear what you did.
- Not very clear which is the aim of the study.
- Please rewrite the abstract.
Introduction
- Page 1 rows 35-38. Please split this sentence because it's too long.
- Page 2 rows 50-51. To my view, other studies that most probably report the opposite should be mentioned. I don’t think that just one study could define the attitude of health personnel.
Methods
- Page 2 row 78. Don’t think that starting methods part with this phrase is correct “Following the example of the authors”. To my view is better to start with something like: According to other similar studies that focused on….
- To my view the terms used for research limit to much the number of studies. Even the term mental health in the title/abstract wasn’t used. Could authors explain why this was done and how this affect the validity of the study?
- What about studies language? Only English written?
- What about study types? All of them were included?
Results
- The section Document types and languages should be rewritten. Is too long and by reading it you lose the meaning.
- Page 6 rows 171-174 should be moved to the limitations part.
Discussion/conclusion
- Which is the added value of this article. What new brings in the international arena. A statement should be done about this.
- Based on the current results, which are some key recommendations that the authors propose?
Reviewer 2 Report
The authors raised an important topic and used appropriate methodology to look for literature within the area, however there is no analysis of results that arise from studies. I understand that authors focused mainly on the statistical analysis of papers, but it would help if the bibliometric analysis is supported with some recommendations and conclusions, even general ones, from those studies. The reasons for the rise in numbers of papers published in recent years should be described. The manuscript is incredibly difficult to read due to extremely poor English grammar, repetitions and lack of clarity, particularly at the beginning. To point out a few examples: within the title it should be “Stigma towards”. Within the abstract, point 2 is unreadable. Sentences are too long and difficult to follow throughout the paper. It is not clear from the start what the authors want to explore. Is it a stigma among nurses against patients with mental disorders or is it about stigmatization of health professionals with mental disorders?
There is a lot of good information related to a number of papers, co-authorships matching and countries and institutions related to publications. Results are nicely presented but the meaningfulness of those results could be better explored. I am struggling to see any significance of this work. I think the benefit from such analysis should really refer to findings and recommendations from all studies. Unfortunately, this manuscript lacks such conclusions.
Round 2
Reviewer 2 Report
The authors provided an improved version of the manuscript with the clearly stated aim. The written language is improved, but not throughout the text. I understand that the authors would like to improve possible collaborations between different groups by providing statistical analysis (bibliometric analysis) of papers in the subject. However, researchers seeking collaborative grants can easily make those links in many other areas. The authors indicate “new lines of research”, but which ones?
Despite nice figures and analysis, I personally struggle to see a deeper significance of this work and I am sorry if my view is disappointing to the authors.
Author Response
Thank you for your comments on the article we have submitted for revision. We hope that the changes we have made are in line with your requests and will help us to improve the quality of the final article. Attached is the corrected manuscript.
Thank you again,
Yours sincerely,
